bioengineering/neuroscience/biomedical engineering

cerebral aneurysm, computational fluid dynamics, hemodynamics, vessel wall enhancement, magnetic resonance imaging

**Author for correspondence:**
Vincent M. Tutino
e-mail: vincentt@buffalo.edu

# Aneurysm risk metrics and hemodynamics are associated with greater vessel wall enhancement in intracranial aneurysms

Sricharan S. Veeturi[1,2], Hamidreza Rajabzadeh-Oghaz[1,4], Nándor K. Pintér[5], Muhammad Waqas[1,4], David M. Hasan[6], Kenneth V. Snyder[1,4], Adnan H. Siddiqui[1,4] and Vincent M. Tutino[1,2,3,5]

[1]Canon Stroke and Vascular Research Center, [2]Department of Mechanical and Aerospace Engineering, [3]Department of Pathology and Anatomical Sciences, and [4]Department of Neurosurgery, University at Buffalo, Buffalo, NY, USA
[5]DENT Neurologic Institute, Buffalo, NY, USA
[6]Department of Neurosurgery, University of Iowa Health Care, Iowa City, IA, USA

SSV, 0000-0002-8114-3817; NKP, 0000-0002-7714-143X

Vessel wall enhancement (VWE) in contrast-enhanced magnetic resonance imaging (MRI) is a potential biomarker for intracranial aneurysm (IA) risk stratification. In this study, we investigated the relationship between VWE features, risk metrics, morphology and hemodynamics in 41 unruptured aneurysms. We reconstructed the IA geometries from MR angiography and mapped pituitary stalk-normalized MRI intensity on the aneurysm surface using an in-house tool. For each case, we calculated the maximum intensity ($CR_{stalk}$) and IA risk (via size and the rupture resemblance score (RRS)). We performed correlation analysis to assess relationships between $CR_{stalk}$ and IA risk metrics (size and RRS), as well as each parameter encompassed in RRS, i.e. aneurysmal size ratio (SR), normalized wall shear stress (WSS) and oscillatory shear index. We found that $CR_{stalk}$ had a strong correlation (Pearson correlation coefficient, PCC = 0.630) with size and a moderate correlation (PCC = 0.472) with RRS, indicating an association between VWE and IA risk. Furthermore, $CR_{stalk}$ had a weak negative correlation with normalized WSS (PCC = −0.320) and a weak positive correlation with SR (PCC = 0.390). Local voxel-based analysis showed only a weak negative correlation between normalized WSS and contrast-enhanced MRI signal

intensity (PCC = −0.240), suggesting that if low-normalized WSS induces enhancement-associated pathobiology, the effect is not localized.

# 1. Introduction

Vessel wall enhancement (VWE) in intracranial aneurysms (IAs) is a potential imaging biomarker for aneurysmal rupture risk. It is a phenomenon observed in contrast-enhanced magnetic resonance imaging (MRI), in which the aneurysm wall exhibits markedly higher signal intensity. The potential utility of VWE in distinguishing stable IAs from unstable and rupture-prone IAs has been demonstrated [1,2], with several groups reporting stable aneurysms tend to have little to no enhancement [3–5]. Robust quantification and standardization of VWE is important for its use as a biomarker. To this end, researchers defined a quantitative metric of IA wall enhancement called the aneurysm-to-pituitary stalk contrast ratio ($CR_{stalk}$), which is the ratio of maximum VWE intensity across the IA to the intensity at the pituitary stalk. Rao *et al.* [6] found that $CR_{stalk}$ delineated larger (and riskier) IAs from smaller IAs, with a sensitivity = 81.5% at a threshold of 0.6. Omodaka *et al.* [5] also found that $CR_{stalk}$ was significantly higher in evolving aneurysms as compared to stable ones. $CR_{stalk}$ has further shown promise as a clinical marker, as it has been shown to be reliably assessed across multiple different types of MRI scanners and across MRI scanners with different imaging strengths.

Aneurysmal wall enhancement has been related to histopathological changes associated with IA progression, namely inflammatory cell infiltration, neovascularization and degeneration of the intima [7,8]. However, the mechanism behind this association remains unknown. One potential explanation is that VWE is related to flow-driven pathological remodelling in the IA sac. Hemodynamics is a key factor that drives IA pathophysiology, playing a critical role in aneurysm initiation, growth and rupture [9]. Studies using three-dimensional imaging and computational fluid dynamics (CFD) have identified and demonstrated the clinical utility of hemodynamic metrics associated with ruptured IAs [10,11]. For example, one such metric developed was rupture resemblance score (RRS), a multivariate logistic regression model based on size ratio (SR), normalized wall shear stress (WSS) and oscillatory shear index (OSI) that could accurately identify ruptured aneurysms with a sensitivity of 90% [12]. A recent study also demonstrated that RRS was also significantly higher in growing IAs compared to those which remained stable, with unstable IAs having lower normalized WSS and higher OSI [13].

In this study, we hypothesized that IA wall enhancement is related to IA risk because it is directly associated with aberrant hemodynamic and morphologic parameters, such as those assessed in the RRS (normalized WSS and OSI). Indeed, preliminary studies have shown that hemodynamic metrics averaged across the entire IA sac, such as IA averaged normalized WSS and OSI, are associated with VWE. For example, Khan *et al.* [14] found sac-averaged MRI signal intensity was related to low IA-averaged normalized WSS. Other reports have subjectively divided the IAs into di- or trichotomized regions and performed a similar analysis. However, in all studies, the lack of means to assess the detailed localization between VWE and flow has limited the understanding of how hemodynamics is spatially related to enhancement. Furthermore, most studies did not use an objective, normalized quantification of VWE, such as $CR_{stalk}$. Thus, we sought to implement an objective co-mapping technique for VWE quantification and visualization to study the local association between $CR_{stalk}$ and aneurysmal hemodynamics through a voxel-based analysis in order to shed light on the spatial relationship between VWE and IA flow.

To this end, we first analysed the relationship between the VWE metric $CR_{stalk}$ and risk assessed by IA size and RRS. Then, we examined if $CR_{stalk}$ was related to the underlying morphological metric (SR) and aneurysm-averaged flow metrics (normalized WSS and OSI) used to calculate RRS. To determine if flow and VWE are spatially related, we performed a voxel-based correlation analysis between the mapped VWE intensity and the hemodynamic variables. This preliminary study works towards understanding the underlying relationship between IA wall enhancement, rupture risk and intra-aneurysmal flow.

# 2. Methods

## 2.1. Patient cohort

This study was approved by the Institutional Review Board at the University at Buffalo (study 00004370). Patient consent was waived for this retrospective analysis. We collected consecutive de-identified

MRA, MRI images and medical history from patients undergoing vessel wall MRI for IA at Dent Neurologic Institute between September 2019 and July 2020. Aneurysms located at the cavernous segment of the internal carotid artery (ICA) were excluded because the high signal intensity of the cavernous sinus on the contrast-enhanced images prevented accurate identification and delineation of the IA wall.

## 2.2. Vessel wall imaging and quantification

Our pipeline for vessel wall imaging and quantification is described in detail in our previous publication [15] and in the electronic supplementary material, Methods. Briefly, we obtained MRA images, and non-enhanced and contrast-enhanced MRI for each patient. We then normalized the contrast-enhanced MRI images using the average intensity value of five randomly sampled points at the pituitary stalk on the contrast-enhanced MRI image [6]. We then used an open source platform (3D Slicer) to register the MRA on the contrast-enhanced MRI image [16]. The MRA images were segmented using a level set technique (http://www.vmtk.org) in order to generate the luminal representation of IA and surrounding vessels. Using an inverse distance-weighted interpolation technique, we mapped the signal intensities onto the luminal surface. This same luminal surface was used for CFD simulations, as described in the following section. To obtain a single enhancement metric for each IA, we calculated the maximum intensity on the sac, or $CR_{stalk}$.

## 2.3. Image-based computational fluid dynamics

We performed CFD to obtain intra-aneurysmal hemodynamics. Segmented MRA images were pre-processed using an open-source software (Meshmixer) before CFD. CFMesh was used to generate uniform polyhedral mesh with a base size of 0.2 mm and four boundary layers to more accurately capture near-wall flow. The number of elements in different geometries ranged from 1.8 to 6.5 million elements. We performed CFD mesh convergence test on a representative case. The mesh converged with polyhedral elements with a base size 0.2 mm. Details of the mesh convergence analysis are available in the electronic supplementary material, data, and are shown in the electronic supplementary material, figure S1.

Since we were unable to collect patient-specific flow data from the subjects, a representative flow waveform from the ICA of a healthy subject was scaled to achieve a cycle averaged velocity of $0.27 \, \mathrm{m \, s^{-1}}$ with a plug profile across the artery cross-section for CFD simulations [17]. For middle cerebral artery (MCA) and anterior communicating (ACom) aneurysms, the waveform was dampened by 30% to factor in the transit from the cervical to the cavernous segments [18]. For IAs located on the basilar artery (BA), we used a constant cycle averaged velocity assumption of $0.328 \, \mathrm{m \, s^{-1}}$ at the BA [19]. We performed transient CFD simulations by convoluting the inflow rates based on location of the inlet with the normalized waveform. A modified Murray's law was used for determining the flow split at each bifurcation in the vasculature, which was based on the diameters of the vessels at each bifurcation [20].

Our CFD simulations were run on the high-performance computing cluster at the University at Buffalo. OpenFOAM (v. 6), an open-source finite volume Navier–Stokes solver, was implemented for all the CFD simulations. The walls of the geometry were assumed to be rigid. Blood was modelled as an incompressible Newtonian fluid with a viscosity of $0.0035 \, \mathrm{N \, s \, m^{-2}}$ and a density of $1060 \, \mathrm{kg \, m^{-3}}$. A PIMPLE algorithm was used for pressure velocity coupling to accommodate the large Courant numbers. A time step of 0.005 s was used and two cardiac cycles were simulated. The results from the second cardiac cycle were averaged to obtain the time-averaged hemodynamic quantities. All data obtained from the CFD simulations were processed using an open-source post-processing platform (ParaView). For all cases, we computed the time-averaged normalized WSS, which is the WSS averaged over the entire cardiac cycle, and OSI. For normalization of the WSS, we used a segment of the parent artery two diameters in length upstream of the aneurysm. These metrics were used, along with aneurysm SR, to calculate RRS for each case. High-risk cases were defined using size and RRS as: IAs with size greater than 5 mm [21], and IAs with an RRS greater than 30% [10,12] in their respective analysis.

We used the segmented MRA image to generate a surface. An in-house MATLAB code (R2019a, The Mathworks, Natick, MA) was used to compute necessary morphological parameters. SR was defined as the ratio of maximum aneurysm size (from the centroid of the neck plane) to the average diameter across the parent artery segment that was used in WSS normalization.

To spatially relate VWE and hemodynamics, we applied a previously developed technique to co-map MRI intensity and hemodynamic variables onto the aneurysm sac [15]. In brief, this method used an inverse distance-weighted interpolation technique to map signal intensities onto the surface of the segmented MRA images that were used for CFD. The output data of the flow simulations were down sampled to be the same size as the MRI voxels in post-processing so that the MRI data and the CFD results were at the same resolution.

## 2.4. Statistical analyses

To assess the relationship between aneurysmal VWE and IA risk or IA-averaged hemodynamic metrics, and between enhancement intensity and flow, we performed correlation analysis. To quantify the degree of correlation, we assessed the Pearson correlation coefficient (PCC) and $p$-value from the Wald test. An absolute $1 \geq |PCC| \geq 0.8$ represented 'very strong' correlation, $0.79 \geq |PCC| \geq 0.6$ represented 'strong' correlation, $0.59 \geq |PCC| \geq 0.4$ represented 'moderate' correlation, $0.39 \geq |PCC| \geq 0.2$ represented 'weak' correlation and $|PCC| < 0.19$ represented a 'very weak' or no correlation [22]. We also performed statistical significance testing between enhancing and non-enhancing cohorts stratified by the value of $CR_{stalk}$ based on a previous validation study [6] ($CR_{stalk} \geq 0.6$ was classified as an enhancing case). For categorical variables, we used Fisher's exact test. For continuous variables, we first evaluated normality using the Shapiro–Wilk test. Normally distributed variables were compared using a Student's $t$-test. Non-normally distributed variables were compared using a Mann–Whitney U-test. A variable was considered significantly different if $p < 0.05$.

# 3. Results

## 3.1. Patient data

We analysed 41 aneurysms from 38 patients. Patient data for all subjects are shown in table 1. The average age of all the patients was 68.4 years with a majority of patients being female (84.2%); 34.2% of the patients were smokers, and 42.1% were hypertensive. Very few patients had a family history of IAs (13.2%), and only three patients (7.9%) had multiple aneurysms. Many of the aneurysms were located at the ICA (41.5%). Clinical risk factors, such as family history and smoking, are well known to be associated with IA development and rupture [23]. However, in this study, the risk metrics we calculated (IA size and RRS) did not consider such factors, which will be examined in future larger studies.

Figure 1 highlights example cases from both the non-enhancing and enhancing groups and shows their raw MRI images, mapped enhancement intensities and simulated hemodynamics. In general,

**Table 1.** Patient characteristics.[a]

| characteristic | value |
|---|---|
| age (average years ± s.d.) | 68.4 ± 12.7 |
| female gender ($n/n_{total}$) | 32/38 (84.2%) |
| smoking ($n/n_{total}$) | 13/38 (34.2%) |
| hypertension ($n/n_{total}$) | 16/38 (42.1%) |
| family history of IA ($n/n_{total}$) | 5/38 (13.2%) |
| patients with multiple IAs ($n/n_{total}$) | 3/38 (7.9%) |
| IA location ($n/n_{total}$) | |
| PCom | 4/41 (9.8%) |
| ACom | 5/41 (12.2%) |
| ICA | 17/41 (41.5%) |
| MCA | 13/41 (31.7%) |
| BA | 2/41 (4.9%) |

[a]Abbreviations: ACom = anterior communicating artery, BA = basilar artery, IA = intracranial aneurysm, ICA = internal carotid artery, MCA = middle cerebral artery, $n$ = number, PCom = posterior communicating artery, s.d. = standard deviation.

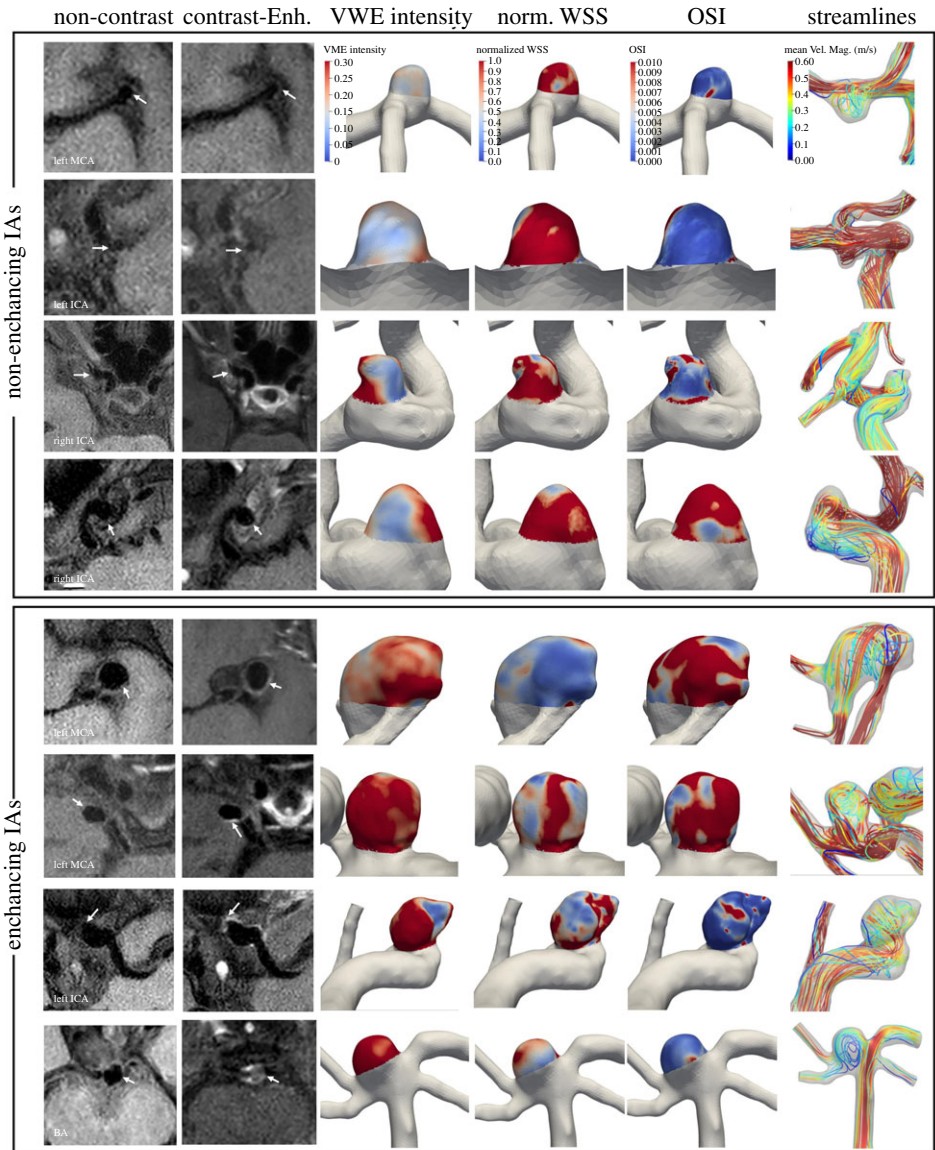

**Figure 1.** Qualitative comparison of VWE intensity and hemodynamics for enhancing and non-enhancing aneurysms. The first two columns show the non-contrast and contrast-enhanced MRI images with the white arrow indicating the location of the aneurysm. We observe that the non-enhancing cases have a higher normalized WSS and a lower OSI, and the enhancing cases have the exact opposite behaviour (lower normalized WSS and higher OSI). (Abbreviations: IA = intracranial aneurysm, Enh = enhanced, norm. WSS = normalized wall shear stress, OSI = oscillatory shear index, Vel. Mag. = velocity magnitude, VWE = vessel wall enhancement, MCA = middle cerebral artery, ICA = internal carotid artery, BA = basilar artery.)

non-enhancing cases had smaller size, higher normalized WSS and lower OSI, while enhancing cases were larger, had lower normalized WSS and higher OSI.

## 3.2. Increased CR$_{stalk}$ is related to larger intracranial aneurysm size and higher rupture resemblance score

Our correlation analysis demonstrated that CR$_{stalk}$ was significantly related to IA risk factors, as quantified by both IA size and RRS. In our data, we found a significant, strong, positive correlation between CR$_{stalk}$ and IA size (PCC = 0.63, $p < 0.001$) (figure 2$a$). When dichotomized based on a 5 mm threshold [21], CR$_{stalk}$ was statistically significantly higher in larger IAs ($p = 0.002$) (figure 2$b$), as exemplified by the cases in figure 2$c$. This also held true when the aneurysms were divided using 7 mm [24] as the size cutoff as shown in the electronic supplementary material, figure S2. Figure 3$a$

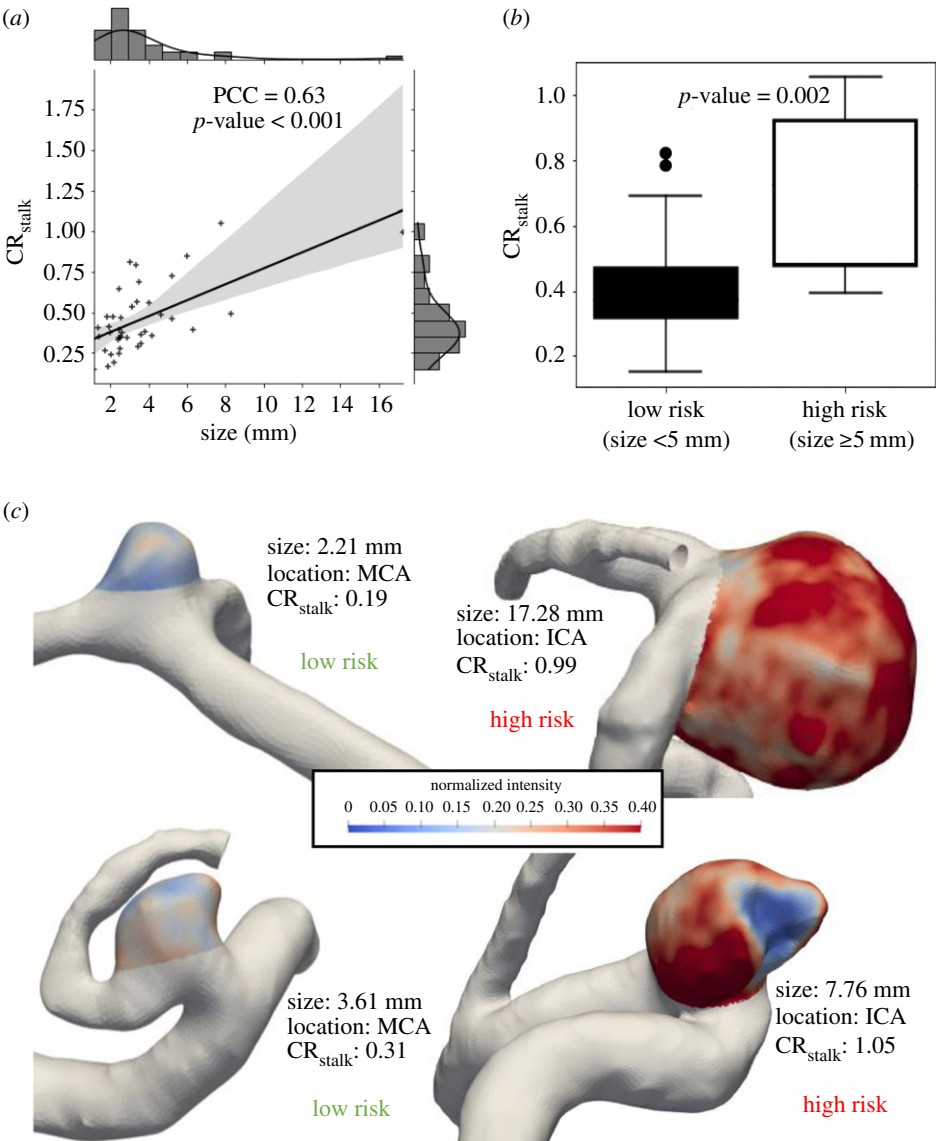

**Figure 2.** Correlation analysis of aneurysm size with $CR_{stalk}$. (*a*) The correlation analysis between $CR_{stalk}$ and size. A histogram of values is displayed opposite to the respective axes and the grey-shaded region represents the 95% confidence interval using 100 bootstrap resamples. (*b*) The high-risk aneurysms defined as aneurysms with size greater than 5 mm have a significantly higher $CR_{stalk}$ value as compared to the low-risk aneurysms. (*c*) A few representative cases of high- and low-risk IAs. (Abbreviations: PPC = Pearson correlation coefficient, $CR_{stalk}$ = aneurysm-to-pituitary stalk contrast ratio, ICA = internal carotid artery, MCA = middle cerebral artery.)

also shows a significant, moderate, positive correlation between $CR_{stalk}$ and RRS (PCC = 0.472, $p = 0.002$). When RRS was dichotomized based on a threshold of 0.30, $CR_{stalk}$ was statistically significantly higher in IAs with greater RRS ($p = 0.004$) (figure 3*b*). This association is demonstrated by the example cases in figure 3*c*. This analysis suggested that while IA enhancement may most strongly be associated with size, it may also be related to aneurysmal hemodynamics, as RRS reflects both normalized WSS and OSI, along with SR.

## 3.3. Aneurysmal size ratio and wall shear stress correlate with maximum enhancement intensity

To further explore the relationship between VWE and RRS, we performed the same analysis between $CR_{stalk}$ and the individual metrics that comprise RRS (SR, normalized WSS and OSI). Regression

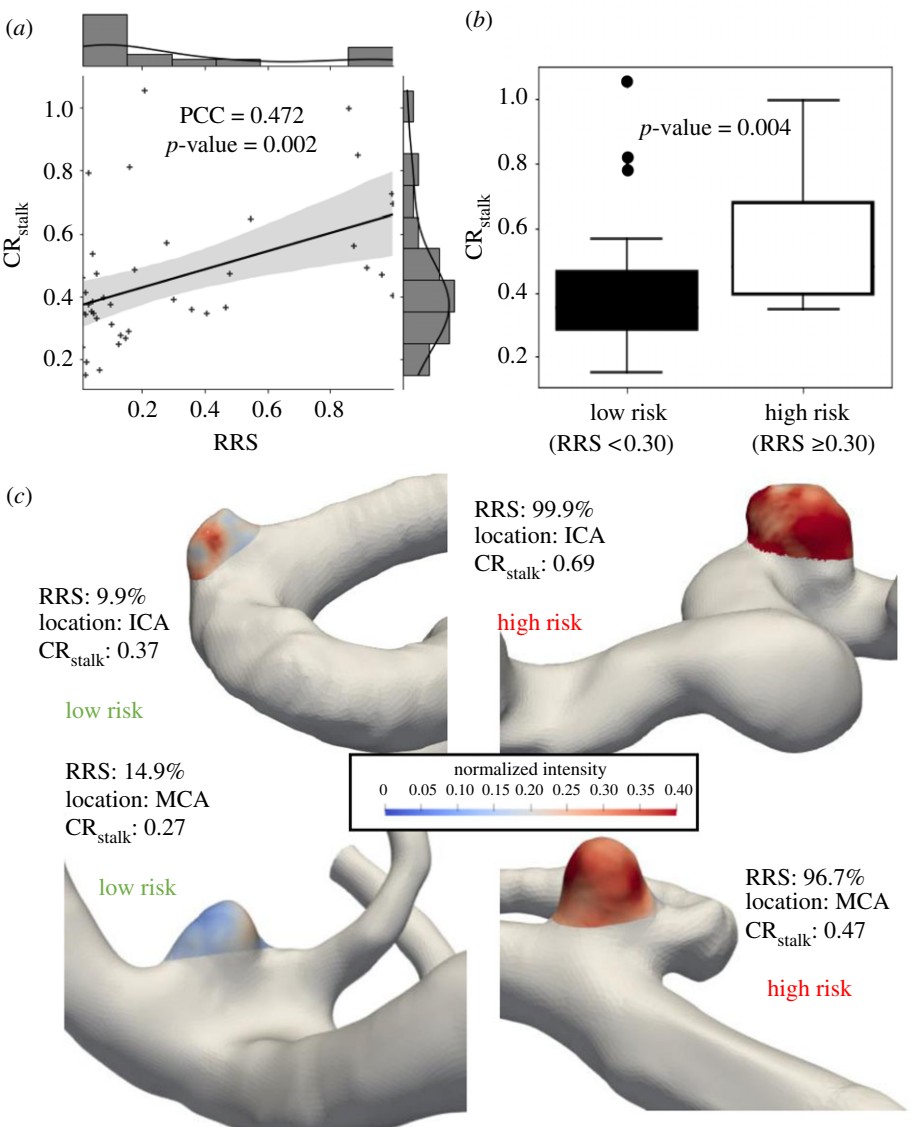

**Figure 3.** Correlation analysis of RRS with $CR_{stalk}$. (*a*) The correlation analysis between $CR_{stalk}$ and RRS. A histogram of values is displayed opposite to the respective axes and the grey-shaded region represents the 95% confidence interval using 100 bootstrap resamples. (*b*) The high-risk aneurysms, defined as aneurysms with RRS > 0.3 have a significantly higher $CR_{stalk}$ value as compared to the low-risk aneurysms. (*c*) A few representative cases of high- and low-risk IAs as defined by RRS. (Abbreviations: PPC = Pearson correlation coefficient, $CR_{stalk}$ = aneurysm-to-pituitary stalk contrast ratio, ICA = internal carotid artery, MCA = middle cerebral artery, RRS = rupture resemblance score.)

analysis showed a significant, weak, positive correlation between $CR_{stalk}$ and SR (PCC = 0.39, $p = 0.012$) (figure 4*a*). When cases were classified as either 'enhancing' or 'non-enhancing' ($CR_{stalk} \geq 0.6$—based on previous analysis demonstrating $CR_{stalk} \geq 0.6$ best delineated enhancing cases [6]), SR was significantly higher in the enhancing group ($p = 0.039$) (figure 4*b*). We also found a significant, weak, negative correlation between $CR_{stalk}$ and normalized WSS (PCC = –0.32, $p = 0.041$) (figure 4*c*). Normalized WSS was also significantly lower in the enhancing group ($p = 0.026$) (figure 4*d*). On the other hand, there was no significant relationship between $CR_{stalk}$ and OSI (PCC = 0.277, $p = 0.08$) (figure 4*e*). However, OSI was significantly higher in enhancing cases as compared to the non-enhancing ones ($p = 0.016$) (figure 4*f*).

Additionally, an IA shape irregularity metric called undulation index (UI; $UI = 1 - (V/V_{ch})$, where $V$ is the volume of the aneurysm and $V_{ch}$ is the volume of the convex hull of the aneurysm) had been shown to be significantly different between ruptured and unruptured IAs in Xiang *et al*. [25]. It was originally considered in the RRS, but was not retained in the final model. Here, we also

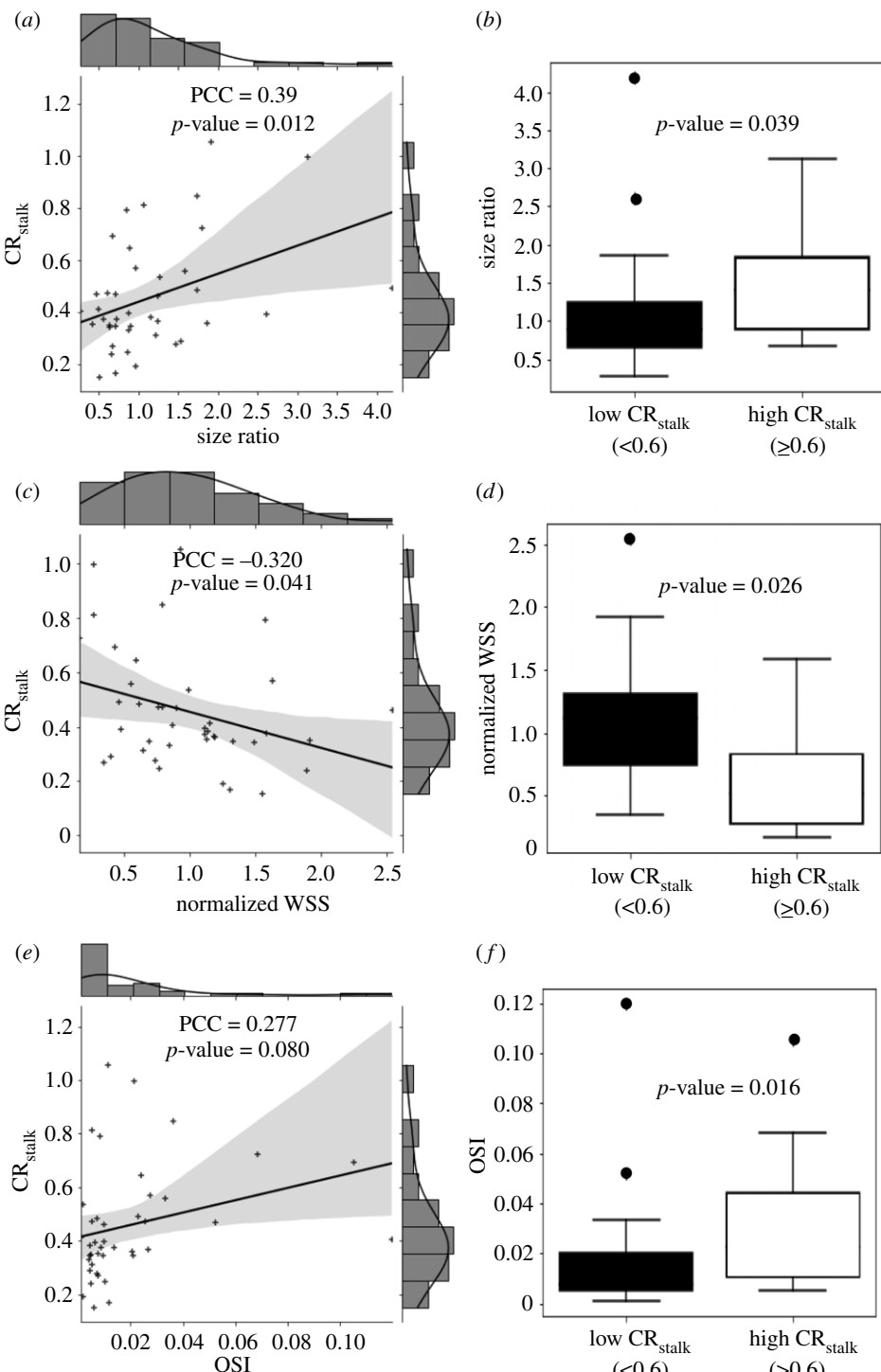

**Figure 4.** Correlation analysis of SR, WSS and OSI with $CR_{stalk}$. (*a,c,e*) Scatter plot of $CR_{stalk}$ and different parameters for all 41 cases and the best linear regression fit therein. We observe that all the parameters have a weak correlation with $CR_{stalk}$. (*b,d,f*) Boxplot of SR, normalized WSS and OSI between enhancing and non-enhancing cases based on maximum $CR_{stalk}$ value ($CR_{stalk} > 0.6$ is defined as an enhancing case). We see that normalized WSS and OSI are significantly different between both cohorts; however, SR is not. (Abbreviations: OSI = oscillatory shear index, PCC = Pearson correlation coefficient, WSS = wall shear stress, $CR_{stalk}$ = aneurysm-to-pituitary stalk contrast ratio.)

evaluated the correlation of $CR_{stalk}$ with UI, as shown in the electronic supplementary material, figure S3, but found no statistically significant correlation or significant difference in UI between enhancing and non-enhancing cases.

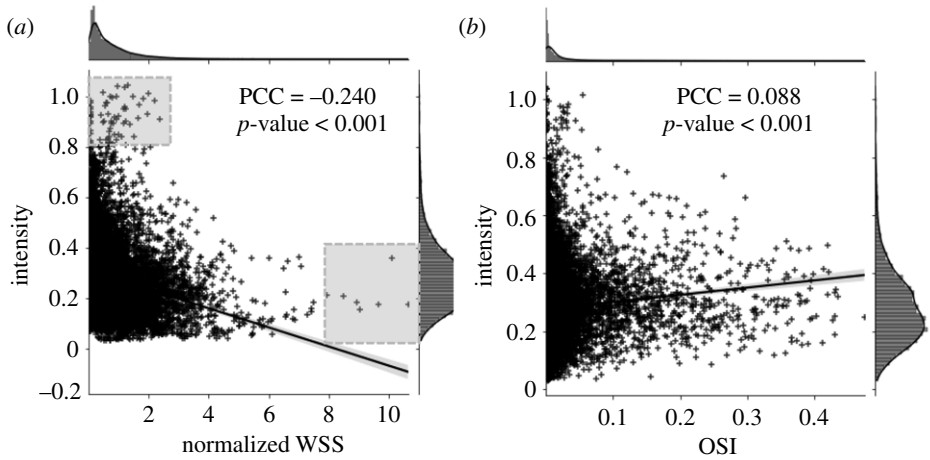

**Figure 5.** Local hemodynamic correlation between mapped intensity and normalized WSS and OSI. This displays a scatter plot of normalized WSS (*a*) and OSI (*b*) against intensity. The extremities, i.e. high-intensity regions and high-normalized WSS regions, are marked in grey boxes. We observe that the normalized WSS has a low negative correlation with intensity and OSI has a very low, almost negligible, correlation with intensity. (Abbreviations: OSI = oscillatory shear index, PCC = Pearson correlation coefficient, WSS = wall shear stress.)

## 3.4. Wall shear stress is weakly correlated with enhancement intensity on a local level

In order to determine if the relationship between aneurysmal hemodynamics and wall enhancement was a localized phenomenon, we co-mapped enhancement intensity and flow from CFD using our previously developed image analysis pipeline [15]. Our voxel-based regression analysis across all cases ($n = 14\,197$ points across all cases) revealed a significant, yet weak, negative correlation between enhancement intensity and normalized WSS (PCC = –0.24, $p < 0.001$) (figure 5*a*). Investigating OSI, we also found a significant, yet very weak, positive correlation with enhancement intensity (PCC = 0.088, $p < 0.001$) (figure 5*b*). Interestingly, in observing the correlation between intensity and normalized WSS, all regions with the highest normalized WSS (greater than 8.0) had exclusively low enhancement (less than 0.4), and all regions with the highest intensity (greater than 0.6) had exclusively low-normalized WSS (less than 3.0) (see shaded regions in figure 5*a*). This trend was not observed for OSI.

# 4. Discussion

In this study, we investigated the relationship between aneurysmal wall enhancement, risk metrics (i.e. size and RRS) and aneurysmal flow. We found a significantly higher maximum normalized contrast intensity among high-risk IAs based on risk assessment via size and RRS. Of the metrics used to calculate RRS, IA SR and sac-averaged normalized wall shear stress (normalized WSS) were significantly associated with the degree of VWE, but sac-averaged OSI was not. Locally, there was a low correlation between enhancement intensity and normalized WSS, although a trend between high-normalized WSS and low enhancement, and high-intensity and low-normalized WSS was apparent.

The first objective of this study was to determine if IA wall enhancement was related to existing metrics of IA risk. From our analysis, we found the $CR_{stalk}$ was significantly related to the surrogate risk metric, IA size. This result is similar to that of Roa *et al.* [6], who also used size (greater than or equal to 7 mm indicated high-risk IAs) as a surrogate for IA risk/instability, and found that $CR_{stalk}$ could delineate high- from low-risk IAs. We also found that $CR_{stalk}$ was significantly related to rupture risk, as assessed by RRS. This is a multivariate classifier of IA rupture, in which higher probability of rupture is associated with larger SR, lower normalized WSS and higher OSI. Xiang *et al.* [12] demonstrated a sensitivity of 90% in classifying aneurysm rupture status using an RRS greater than 0.3. However, based on our analysis, the relationship between enhancement and RRS was substantially weaker than that of $CR_{stalk}$ and IA size. These results suggest an underlying relationship between aneurysm morphology (size) and IA enhancement, and to a lesser degree, a relationship between aneurysm hemodynamics and VWE.

Morphologically, our correlation analysis showed that aneurysmal VWE was most strongly associated with aneurysm size and moderately with SR. These results were not surprising, as IA size has been

associated with IA VWE across multiple analyses [26,27]. For example, Backes *et al.* [26] found that the strongest determinant of aneurysm wall enhancement in 89 unruptured IAs was IA size. This was later confirmed by Liu *et al.* [27] who found size was independently associated with enhancement in a cohort of 61 IAs. It has been hypothesized that the reason for the relationship between size and enhancement is the presence of vasa vasorum in larger aneurysms [28]. Vasa vasorum has been associated with VWE in the histological analysis [7,29,30], and studies have found that aneurysms larger than 4 mm exhibit the presence of vasa vasorum in the adventitia [28,31]. Therefore, as IA size increases, one may observe an increase in vasa vasorum and consequentially greater VWE.

Aside from IA geometry, a moderate correlation between $CR_{stalk}$ and RRS also suggested an association between enhancement and aneurysmal flow. Our data showed that sac-averaged normalized WSS was weakly correlated to $CR_{stalk}$. Additionally, enhancing IAs had significantly lower normalized WSS and significantly higher OSI. These results mirror those of other studies in the scientific literature [14,32,33]. For example, Khan *et al.* found that enhancing IA cases had lower normalized WSS and velocity, but no statistically significant difference in OSI from IAs that were not enhancing. Lv *et al.* [33] also broadly classified the VWE status of IAs (non-enhancing, partial enhancement and circumferential enhancement) and found that aneurysms displaying enhancement features had lower normalized WSS, higher low shear area and higher relative residence time (RRT) of blood. Aneurysmal VWE may be related to low WSS because the aberrantly low flow has been shown to elicit an inflammatory response in the endothelium, leading to a leaky intima and perpetuating the recruitment and extravasation of inflammatory cells and their subsequent degeneration of the media, akin to the pathogenesis of atherosclerosis [34]. Indeed, these biological features have also been observed in the walls of enhancing IAs on histological analyses [7,29]. Therefore, in addition to the potential influence of vasa vasorum in larger IAs, our results may also indicate low WSS-driven pathological vascular remodelling could also propagate VWE through the presence of inflammation in the IA wall [34].

While Khan *et al.* and Lv *et al.* explored the relationship of sac-averaged hemodynamics and VWE, others have taken a more local approach. To further investigate the relationship between flow and VWE, several groups have measured this relationship by identifying hemodynamic differences in enhancing and non-enhancing regions in IAs. Xiao *et al.* [35] manually segmented enhancing areas and found low WSS and low OSI were associated with VWE. Similarly, Larsen *et al.* [36] quantified the ratio of enhancing regions in IA sacs, but found that low WSS and maximum OSI, as well as low shear area were associated with higher VWE. Recently, Hadad *et al.* [32] also identified enhancing regions via manual segmentation and reported average WSS and WSS divergence were lower where there was enhancement. Taking a similar approach, we used our co-mapping technique to perform a voxel-based correlation analysis for more accurate investigation of the relationship between flow and VWE. Based on our data, however, we observed only a weak relationship between normalized WSS and VWE, and no correlation between OSI and VWE. This suggests that there is only a weak local relationship between WSS and enhancement, or that the downstream biological effects of low WSS (those related to VWE) may diffuse through the IA tissue over time, and thus not be highly localized to patterns of flow.

The contrast uptake into the wall may depend on several factors, including wall permeability, the amount of contrast in the blood and the blood's residence time in the IA. While WSS can affect the permeability of the endothelium, other parameters, such as RRT, which is higher in larger IAs, may also be related to VWE features of an IA. We calculated RRT from the WSS and OSI from our cases and found that it was indeed significantly different between enhancing and non-enhancing IAs. However, we found no significant correlation between $CR_{stalk}$ and RRT on both an IA-averaged and a local voxel-based level (see electronic supplementary material, figure S4). These results support WSS as the major, albeit not local, hemodynamic correlate of VWE in IAs.

This study has several limitations. First, we could only relate $CR_{stalk}$ to existing metrics that have been previously shown to differentiate ruptured versus unruptured IAs. Longitudinal data are required to truly determine the relationship between VWE parameters and aneurysm instability, by tracking those IAs that grow and/or rupture. Second, the small sample size of this study may limit our confidence in the results. Future studies with larger cohorts are required. Third, we adopted several commonly used assumptions for our CFD simulations. Due to a lack of patient-specific flow information, we assumed a generic inlet waveform and a constant, location-based inlet flow rate. The time-averaged inlet velocities were assumed to be the same across different patients, the walls were assumed to be rigid, and blood was assumed to be a Newtonian fluid. However, in a recent study, it was demonstrated that WSS and normalized WSS were not greatly affected by varying boundary conditions (including outlet boundary conditions, and inlet velocity and pulsatility), whereas OSI was [37]. Therefore, there may be unknown degrees of error in certain hemodynamic values, particularly

OSI. Patient-specific flow conditions are needed for more accurate CFD simulations in the future. Fourth, we did not consider clotting that can occur at various stages during IA development in the IA sac (although none was noted on black blood images), which could affect the VWE of the wall. Finally, down-sampling the CFD data to maintain the same resolution as the MRI images in our voxel-based correlation analysis could lead to data loss in our flow simulations. In the future, MRI at higher spatial resolutions could help facilitate more accurate local comparisons of VWE and IA flow and better detection of other wall phenomena such as intraluminal thrombus formation.

# 5. Conclusion

This preliminary study shows that aneurysmal $CR_{stalk}$ is associated with IA risk factors as evaluated by both aneurysm size and RRS. The presence of VWE was associated primarily with IA size and, to a lesser degree, SR and lower sac-averaged WSS. There was no relationship between $CR_{stalk}$ and OSI. The spatial association between WSS and MRI signal intensity was generally poor, although regions of highest intensity were located exclusively at areas of lower normalized WSS. Based on these findings, we suspect that low flow creates a favourable environment in the vessel wall for VWE, but this phenomenon is not a local one.

Ethics. This study was approved by the institutional review board (IRB) at the University at Buffalo (Study 00004370).
Data accessibility. Data are available for all the RRS, size, $CR_{stalk}$ and hemodynamic values used to plot the figures shown in the paper.
    The data are provided in the electronic supplementary material [38].
Authors' contributions. Study design: S.S.V., H.R.O. and V.M.T. Data collection: S.S.V., N.P., A.H.S. and K.V.S. Analysis: S.S.V. and V.M.T. Writing the manuscript all authors. Funding: K.V.S., A.H.S. and V.M.T.
Disclosures/competing interests. S.S.V.: none. H.R.O.: none. N.K.P.: none. M.W.: none. D.M.H.: none. K.V.S.: consulting/ teaching: Canon Medical Systems Corporation, Penumbra Inc., Medtronic, Jacobs Institute. Co-founder: Neurovascular Diagnostics, Inc. A.H.S.: financial interest/investor/stock options/ownership: Amnis Therapeutics, Apama Medical, BlinkTBI, Inc., Buffalo Technology Partners, Inc., Cardinal Health, Cerebrotech Medical Systems, Inc., Claret Medical, Cognition Medical, Endostream Medical, Ltd, Imperative Care, International Medical Distribution Partners, Rebound Therapeutics Corp., Silk Road Medical, StimMed, Synchron, Three Rivers Medical, Inc., Viseon Spine, Inc. Consultant/advisory board: Amnis Therapeutics, Boston Scientific, Canon Medical Systems USA, Inc., Cerebrotech Medical Systems, Inc., Cerenovus, Claret Medical, Corindus, Inc., Endostream Medical, Ltd, Guidepoint Global Consulting, Imperative Care, Integra, Medtronic, MicroVention, Northwest University—DSMB Chair for HEAT Trial, Penumbra, Rapid Medical, Rebound Therapeutics Corp., Silk Road Medical, StimMed, Stryker, Three Rivers Medical, Inc., VasSol, W.L. Gore & Associates. National PI/steering committees: Cerenovus LARGE Trial and ARISE II Trial, Medtronic SWIFT PRIME and SWIFT DIRECT Trials, MicroVention FRED Trial & CONFIDENCE Study, MUSC POSITIVE Trial, Penumbra 3D Separator Trial, COMPASS Trial, INVEST Trial. Principal investigator: Cummings Foundation grant. V.M.T.: principal investigator: National Science Foundation Award no. 1746694, NIH NINDS award R43 NS115314-0, Clinical and Translational Science Institute grant. Co-founder: Neurovascular Diagnostics, Inc.
Funding. This study was funded by SUNY Research Seed Grant (grant no. RSG201049.2), Brain Aneurysm Foundation and National Institutes of Health (grant no. 1R43NS115314-01).
Acknowledgements. We would like to thank the University at Buffalo's Center for Computational Research at the University at Buffalo for providing computational resources for our CFD simulations.

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
