## [Peer Review File · Royal Society Open Science]

Review History

RSOS-211119.R0 (Original submission)

Review form: Reviewer 1

Is the manuscript scientifically sound in its present form?

Yes

Are the interpretations and conclusions justified by the results?

Yes

Is the language acceptable?

Yes

Do you have any ethical concerns with this paper?

No

Have you any concerns about statistical analyses in this paper?

No

Recommendation?

Accept with minor revision (please list in comments)

Comments to the Author(s)

The revised manuscript is stronger and provides important details on imaging and modeling methods used in the study. Most of my concerns were addressed; the voxel-based analysis of the relation between wall enhancement and relative residence time is particularly appreciated.

I have only a few minor suggestions and edits listed below:

Page 4, line 12: "Indeed, preliminary studies have shown that hemodynamics averaged across the entire IA sac.." please change to "hemodynamics to "hemodynamic metrics", as hemodynamics is a science (e.g. physics, chemistry) rather than a list of aneurysm characteristics. (In fact, I could argue that the title of this paper has the same issue).

Page 6, line 24: "Pimple was used.. " I would suggest "Pimple algorithm was used"

Page 6, line 31: "All the data obtained from the CFD simulations was processed" -- data is plural, please change to "were used"

Page 6, line 48-52: "Size ratio was defined as the maximum size of the aneurysm from the centroid of the neck plane to the ratio of average diameter across the parent artery segment used for WSS normalization." This sounds a bit confusing, perhaps could be rephrased (e.g., remove the word ratio before the average diameter?)

Page 7, line 14: "To assess the relationship between aneurysmal VWE, and IA risk or IA-averaged hemodynamics" again, hemodynamics is a science, not a list of variables and this cannot be averaged. Please use "hemodynamic variables" or "metrics"

Page 8, line 18: "Our correlation analysis demonstrated that CRstalk was significantly related to IA risk, as quantified by both IA size and RRS" -- as discussed in the review of the original manuscript, the analysis demonstrated that CRstalk was related to IA risk factors, not IA risk (we do not know whether the aneurysms actually proceeded to grow and rupture). I recommend that this is corrected by just adding the word "factors" after "IA risk"

Page 9 line 10: please define undulation index for readers not familiar with the commonly used aneurysm risk metrics

Page 9, line 53: "From RRS, aneurysm size ratio (SR) and sac-averaged normalized wall shear stress (normalized WSS), were also significantly different between enhancing and non-enhancing cases, but sac-averaged oscillatory shear index (OSI) was not. -- Based on RRS analysis?"

Page 12, line 27-28: "These results support WSS as the major, albeit not local, hemodynamic correlate of VWE in IAs. of" -- please remove "of"

Page 12, lines 31-33: "First, we could only relate CRstalk to existing metrics that have been primarily shown to differentiate ruptured vs. unruptured IAs" -- I would say "previously" rather than "primarily"

Page 12, line 44-45: "The inlet velocities were assumed to be constant at the inlets" -- this is not correct, as use prescribed a pulsatile waveform at the inlets.

Page 13, line 21 (Conclusion): "This preliminary study shows that aneurysmal CRstalk is association with IA risk as measured by both aneurysm size and the RRS" -- it should be "is associated with", not "association". Also, please change to "IA risk factors" instead of "IA risk", as discussed above.

Review form: Reviewer 2

Is the manuscript scientifically sound in its present form?

Yes

Are the interpretations and conclusions justified by the results?

Yes

Is the language acceptable?

Yes

Do you have any ethical concerns with this paper?

No

Have you any concerns about statistical analyses in this paper?

No

Recommendation?

Accept as is

Comments to the Author(s)

No new comments

Decision letter (RSOS-211119.R0)

Dear Mr Veeturi

On behalf of the Editors, we are pleased to inform you that your Manuscript RSOS-211119 "Aneurysm Risk Metrics and Hemodynamics are Associated with Greater Vessel Wall Enhancement in Intracranial Aneurysms" has been accepted for publication in Royal Society Open Science subject to minor revision in accordance with the referees' reports. Please find the referees' comments along with any feedback from the Editors below my signature.

Please also confirm an active email address for your colleague with the email address hrajabza@buffalo.edu or ask them to ensure that emails from our office are 'white listed', as messages to their address are currently bouncing.

Please submit your revised manuscript and required files (see below) no later than 7 days from today's (ie 02-Sep-2021) date. Note: the ScholarOne system will 'lock' if submission of the revision is attempted 7 or more days after the deadline. If you do not think you will be able to meet this deadline please contact the editorial office immediately.

on behalf of Prof Pietro Cicuta (Subject Editor)
openscience@royalsociety.org

Associate Editor Comments to Author:

Thank you for taking such care and attention with the comments from the earlier review at JRSI. We're pleased that the reviewers are broadly satisfied with the changes you have made, but a number of remaining modifications are required before the paper may be accepted for publication - please ensure you carefully respond to and incorporate these remaining matters before resubmitting.

Reviewer comments to Author:

Reviewer: 1

Comments to the Author(s)

The revised manuscript is stronger and provides important details on imaging and modeling methods used in the study. Most of my concerns were addressed; the voxel-based analysis of the relation between wall enhancement and relative residence time is particularly appreciated.

I have only a few minor suggestions and edits listed below:

Page 4, line 12: "Indeed, preliminary studies have shown that hemodynamics averaged across the entire IA sac.." please change to "hemodynamics to "hemodynamic metrics", as hemodynamics is a science (e.g. physics, chemistry) rather than a list of aneurysm characteristics. (In fact, I could argue that the title of this paper has the same issue).

Page 6, line 24: "Pimple was used.." I would suggest "Pimple algorithm was used"

Page 6, line 31: "All the data obtained from the CFD simulations was processed" -- data is plural, please change to "were used"

Page 6, line 48-52: "Size ratio was defined as the maximum size of the aneurysm from the centroid of the neck plane to the ratio of average diameter across the parent artery segment used for WSS normalization." This sounds a bit confusing, perhaps could be rephrased (e.g., remove the word ratio before the average diameter?)

Page 7, line 14: "To assess the relationship between aneurysmal VWE, and IA risk or IA-averaged hemodynamics" again, hemodynamics is a science, not a list of variables and this cannot be averaged. Please use "hemodynamic variables" or "metrics"

Page 8, line 18: "Our correlation analysis demonstrated that CRstalk was significantly related to IA risk, as quantified by both IA size and RRS" -- as discussed in the review of the original manuscript, the analysis demonstrated that CRstalk was related to IA risk factors, not IA risk (we do not know whether the aneurysms actually proceeded to grow and rupture). I recommend that this is corrected by just adding the word "factors" after "IA risk"

Page 9 line 10: please define undulation index for readers not familiar with the commonly used aneurysm risk metrics

Page 9, line 53: "From RRS, aneurysm size ratio (SR) and sac-averaged normalized wall shear stress (normalized WSS), were also significantly different between enhancing and non-enhancing cases, but sac-averaged oscillatory shear index (OSI) was not. -- Based on RRS analysis?"

Page 12, line 27-28: "These results support WSS as the major, albeit not local, hemodynamic correlate of VWE in IAs. of" -- please remove "of"

Page 12, lines 31-33: "First, we could only relate CRstalk to existing metrics that have been primarily shown to differentiate ruptured vs. unruptured IAs" -- I would say "previously" rather than "primarily"

Page 12, line 44-45: "The inlet velocities were assumed to be constant at the inlets" -- this is not correct, as we prescribed a pulsatile waveform at the inlets.

Page 13, line 21 (Conclusion): "This preliminary study shows that aneurysmal CRstalk is associated with IA risk as measured by both aneurysm size and the RRS" -- it should be "is associated with", not "association". Also, please change to "IA risk factors" instead of "IA risk", as discussed above.

Reviewer: 2
Comments to the Author(s)
No new comments

===PREPARING YOUR MANUSCRIPT===

Your revised paper should include the changes requested by the referees and Editors of your manuscript. You should provide two versions of this manuscript and both versions must be provided in an editable format:
one version identifying all the changes that have been made (for instance, in coloured highlight, in bold text, or tracked changes);
a 'clean' version of the new manuscript that incorporates the changes made, but does not highlight them. This version will be used for typesetting.

===PREPARING YOUR REVISION IN SCHOLARONE===

- If you are providing image files for potential cover images, please upload these at this step, and inform the editorial office you have done so. You must hold the copyright to any image provided.
- A copy of your point-by-point response to referees and Editors. This will expedite the preparation of your proof.

- Ensure that your data access statement meets the requirements at <https://royalsociety.org/journals/authors/author-guidelines/#data>. You should ensure that you cite the dataset in your reference list. If you have deposited data etc in the Dryad repository, please only include the 'For publication' link at this stage. You should remove the 'For review' link.
- If you are requesting an article processing charge waiver, you must select the relevant waiver option (if requesting a discretionary waiver, the form should have been uploaded at Step 3 'File upload' above).
- If you have uploaded ESM files, please ensure you follow the guidance at <https://royalsociety.org/journals/authors/author-guidelines/#supplementary-material> to include a suitable title and informative caption. An example of appropriate titling and captioning may be found at https://figshare.com/articles/Table_S2_from_Is_there_a_trade-off_between_peak_performance_and_performance_breadth_across_temperatures_for_aerobic_scorpions_in_teleost_fishes_/3843624.

Author's Response to Decision Letter for (RSOS-211119.R0)

See Appendix A.

Decision letter (RSOS-211119.R1)

Dear Mr Veeturi,

I am pleased to inform you that your manuscript entitled "Aneurysm Risk Metrics and Hemodynamics are Associated with Greater Vessel Wall Enhancement in Intracranial Aneurysms" is now accepted for publication in Royal Society Open Science.

on behalf of Pietro Cicuta (Subject Editor)
openscience@royalsociety.org

Appendix A

Reviewer comments to Author:

Reviewer: 1

Comments to the Author(s)

The revised manuscript is stronger and provides important details on imaging and modeling methods used in the study. Most of my concerns were addressed; the voxel-based analysis of the relation between wall enhancement and relative residence time is particularly appreciated.

I have only a few minor suggestions and edits listed below:

Page 4, line 12: "Indeed, preliminary studies have shown that hemodynamics averaged across the entire IA sac.." please change to "hemodynamics to "hemodynamic metrics", as hemodynamics is a science (e.g. physics, chemistry) rather than a list of aneurysm characteristics. (In fact, I could argue that the title of this paper has the same issue).

Authors: We have now made this correction.

Page 6, line 24: "Pimple was used.. " I would suggest "Pimple algorithm was used"

Authors: This has been corrected.

Page 6, line 31: "All the data obtained from the CFD simulations was processed" -- data is plural, please change to "were used"

Authors: We have now made this correction.

Page 6, line 48-52: "Size ratio was defined as the maximum size of the aneurysm from the centroid of the neck plane to the ratio of average diameter across the parent artery segment used for WSS normalization." This sounds a bit confusing, perhaps could be rephrased (e.g., remove the word ratio before the average diameter?)

Authors: Thank you for this suggestion. We have made the suggested edit.

Page 7, line 14: "To assess the relationship between aneurysmal VWE, and IA risk or IA-averaged hemodynamics" again, hemodynamics is a science, not a list of variables and this cannot be averaged. Please use "hemodynamic variables" or "metrics"

Authors: This has been corrected.

Page 8, line 18: "Our correlation analysis demonstrated that CRstalk was significantly related to IA risk, as quantified by both IA size and RRS" -- as discussed in the review of the original manuscript, the analysis demonstrated that CRstalk was related to IA risk factors, not IA risk (we do not know whether the aneurysms actually proceeded to grow and rupture). I recommend that this is corrected by just adding the word "factors" after "IA risk"

Authors: We have now made this edit.

Page 9 line 10: please define undulation index for readers not familiar with the commonly used aneurysm risk metrics

Authors: Thank you for this suggestion. We have now included this on page 9.

Page 9, line 53: "From RRS, aneurysm size ratio (SR) and sac-averaged normalized wall shear stress (normalized WSS), were also significantly different between enhancing and non-enhancing cases, but sac-averaged oscillatory shear index (OSI) was not. -- Based on RRS analysis?"

Authors: Sorry for this confusion. We have now rewritten this sentence to be clearer.

Page 12, line 27-28: "These results support WSS as the major, albeit not local, hemodynamic correlate of VWE in IAs. of" -- please remove "of"

Authors: We have corrected this error.

Page 12, lines 31-33: "First, we could only relate CRstalk to existing metrics that have been primarily shown to differentiate ruptured vs. unruptured IAs" -- I would say "previously" rather than "primarily"

Authors: We have now made this edit.

Page 12, line 44-45: "The inlet velocities were assumed to be constant at the inlets" -- this is not correct, as you prescribed a pulsatile waveform at the inlets.

Authors: You are correct. We have now clarified this - it was applied constantly across all the cases.

Page 13, line 21 (Conclusion): "This preliminary study shows that aneurysmal CRstalk is association with IA risk as measured by both aneurysm size and the RRS" -- it should be "is associated with", not "association". Also, please change to "IA risk factors" instead of "IA risk", as discussed above.

Authors: Thank you for this suggestion. We have made the suggested edit.

Reviewer: 2

Comments to the Author(s)

No new comments